# Various FDM Mechanisms Used in the Fabrication of Continuous-Fiber Reinforced Composites: A Review

**DOI:** 10.3390/polym16060831

**Published:** 2024-03-18

**Authors:** Armin Karimi, Davood Rahmatabadi, Mostafa Baghani

**Affiliations:** 1School of Mechanical Engineering, College of Engineering, University of Tehran, Tehran 1417614411, Irand.rahmatabadi@ut.ac.ir (D.R.); 2Department of Aerospace Engineering, Sharif University of Technology, Tehran 1411713114, Iran

**Keywords:** additive manufacturing, Fused Depositing Modeling (FDM), discontinuous and continuous fibers, reinforcing fibers, mechanisms, in-situ fusion, ex-situ prepreg

## Abstract

Fused Deposition Modeling (FDM) is an additive manufacturing technology that has emerged as a promising technique for fabricating 3D printed polymers. It has gained attention recently due to its ease of use, efficiency, low cost, and safety. However, 3D-printed FDM components lack sufficient strength compared to those made using conventional manufacturing methods. This low strength can be mainly attributed to high porosity and low sinterability of layers and then to the characteristics of the polymer used in the FDM process or the FDM process itself. Regarding polymer characteristics, there are two main types of reinforcing fibers: discontinuous (short) and continuous. Continuous-fiber reinforced composites are becoming popular in various industries due to their excellent mechanical properties. Since continuous reinforcing fibers have a more positive effect on increasing the strength of printed parts, this article focuses primarily on continuous long fibers. In addition to polymer characteristics, different mechanisms have been developed and introduced to address the issue of insufficient strength in 3D-printed FDM parts. This article comprehensively explains two main FDM mechanisms: in-situ fusion and ex-situ prepreg. It also provides relevant examples of these mechanisms using different reinforcing elements. Additionally, some other less frequently utilized mechanisms are discussed. Each mechanism has its own advantages and disadvantages, indicating that further development and modification are needed to increase the strength of 3D-printed FDM parts to be comparable to those produced using traditional methods.

## 1. Introduction

Additive manufacturing, also known as 3D printing or rapid prototyping, is a technique used to manufacture physical components based on computer-aided design (CAD) models, ranging from simple to complex. This is achieved through the layer-by-layer deposition of material. Although this method is not new, its recent popularity among researchers, manufacturers, and hobbyists can be attributed to its time and cost-effectiveness, minimal material waste, reduced emissions, and ability to fabricate various industrial and medical equipment [1,2,3,4,5]. There are several 3D printing processes available, each suitable for different materials. These include Stereolightography (SLA), the world’s first 3D printing innovation used for printing photopolymers, as well as Selective Laser Sintering (SLS) for polymers, Selective Laser Melting (SLM) and Direct Metal Deposition (DMD) for metals, and Fused Deposition Modeling (FDM) for thermoplastics, among others [6,7,8,9,10]. FDM, one of the earliest 3D printing processes, has become widely used for fabricating various functional components and prototypes using engineering thermoplastics. This is because of its safety, durability, and efficiency [11,12,13]. The process works by fabricating a 3D geometry layer by layer, depositing an extruded plastic filament from a nozzle [14,15]. 3D printing filament is the thermoplastic feedstock used in FDM 3D printers. There are many types of filament available, each with different properties and coming in a range of diameters. Filaments consist of a continuous, slender plastic thread that is spooled onto a reel [15].

As shown in Figure 1, the FDM process is simple. It begins by drawing a plastic filament into a liquefier head using drive wheels and heating it until it reaches a semi-liquid state. The process concludes by extruding the semi-liquid material through a nozzle and depositing it on a platform to create a 3D object within a temperature-controlled chamber [16,17,18].

In addition to its functional and efficient application in prototyping, the simplicity and lower cost of FDM’s raw materials have also attracted the attention of researchers and manufacturers for use in other areas. These include developing new materials, biomedical and tissue engineering applications, as well as tooling [14,19,20,21]. However, the process has some shortcomings that limit its broader functional applications. For example, FDM 3D-printed parts have weaker mechanical properties and are prone to delamination between layers, leading to premature failure compared to parts made through traditional plastic injection molding [22,23,24]. These weaknesses may be due to the low strength of the filaments used and/or the FDM process itself [25,26,27]. Figure 2 summarizes the advantages and disadvantages of the FDM process.

3D-printed pure polymers often lack sufficient strength to be used as functional engineering components. As a result, the broader applications of such materials are limited [28]. Therefore, many researchers have recently made efforts to overcome the problem of poor mechanical properties in FDM-printed thermoplastic composites and structural parts. One approach is to add reinforcing elements or fibers (reinforcements) into the filaments [25,29]. These reinforcing materials, mainly fibers, are developed to be added to the base polymer matrix during 3D printing, resulting in the fabrication of 3D components with enhanced mechanical behavior. These reinforcing materials can be divided into continuous and discontinuous (short) fibers [30]. Continuous fibers have a long aspect ratio and generally have a preferred orientation, while short fibers have a short aspect ratio and are produced with a random orientation [28,31]. Composites fabricated with continuous fibers exhibit higher strength than those with short fibers, thanks to their orientation. Therefore, continuous fibers are now becoming more popular than short fibers [32,33,34].

In addition to the strength of the used filament, which affects the mechanical behavior of the resulting composite, the FDM process mechanism also impacts the strength of the product. There are two main mechanisms used in FDM for continuous fiber reinforced composites, each with their own advantages and disadvantages. These include the dual extruder mechanism or ex-situ prepreg method (which uses one nozzle for depositing neat polymer filament and another for printing continuous reinforced fiber), and the in-situ fusion or simultaneous impregnation of dry fiber and matrix in the modified nozzle during printing [6,35,36,37,38]. There are also other less frequently used mechanisms, such as 3D compaction printing, which are modified and developed versions of the two main mechanisms [18,28].

This review paper focuses on the mechanisms and reinforcing materials used in the FDM process. Furthermore, since there have been many comprehensive studies conducted on the FDM of short fibers, but the FDM of continuous fibers has not been extensively studied yet and has the potential to bring higher strength to the composites, this review article places more emphasis on continuous fibers.

## 2. FDM Filaments

### 2.1. Filaments Production and Reinforcing Process

Since filament is a vital part of FDM printing, it is important to understand the production process of filament and how it is reinforced. The process of filament fabrication can be broken down into four steps, starting with raw materials and ending with a spool. The first step in filament production is manufacturing the plastic. Crude oil is heated in an industrial furnace during refinement, separating its various components. Naphtha, a key component in making plastics, is produced during this process. Naphtha, along with catalysts and other chemical components, is combined in a polymerization reactor. The resulting polymerized naphtha products are then compounded and processed by melting and mixing them with other materials to create plastic. The plastic is then granulated into small pieces known as pellets or resin. The second step involves preparing the pellets for shaping. The pellets are placed in an industrial blender and mixed with additives to create a consistent blend with specific properties. Additives can include colorants for color or other materials to enhance properties like impact resistance, strength, and structural integrity. Exotic filaments, such as wood filaments, are made by mixing special additives like sawdust or wood particles with the plastic pellets. Once properly mixed, the pellets undergo a drying phase to remove any absorbed moisture before moving on to the next step. The third step is shaping the pellets into a string-like form through a process of heating and cooling. The pellets are fed into a filament extruder with a heating chamber where they are melted into a gooey substance for easy shaping. The melted pellets are then shaped into a consistent, stranded material known as filament, which is extruded through a nozzle and cooled in water chambers to solidify into its final shape. After cooling, the filament is pulled through water chambers to achieve the desired diameter. The speed at which the filament is pulled determines its diameter, with slower speeds resulting in larger diameters and faster speeds resulting in smaller diameters. Once the filament reaches the correct diameter, it is spooled and measured before being cut and secured. This process continues until the batch of filament is completed [39,40].

### 2.2. Filaments Types

There are various types of 3D printer filaments available. In this article, we will discuss some of the most commonly used types, detailing their mechanical properties, characteristics, advantages, and disadvantages.

#### 2.2.1. Poly (Lactic Acid) (PLA)

PLA is a thermoplastic monomer derived from organic sources, unlike other 3D printer filament types that are produced from petroleum products. PLA is known for being easy to print and environmentally friendly. However, it is brittle and lacks UV resistance. PLA is resistant to warping during printing and is not soluble in water, but it can be dissolved in acetone, methyl ethyl ketone, or caustic soda. Additionally, PLA is considered food safe [41].

#### 2.2.2. Acrylonitrile Butadiene Styrene (ABS)

ABS is a widely used engineering plastic and 3D printing filament type. It exhibits excellent toughness and can withstand relatively high temperatures. Printing with ABS requires high temperatures for both the hot end and the printer bed, as well as heated build volumes for good results. However, all types of ABS tend to warp during printing, leading to poor dimensional accuracy. Despite this, ABS has excellent resistance to wear and tear, making it both tough and impact resistant. ABS is not soluble in water, but organic solvents such as acetone, methyl ethyl ketone, and esters can dissolve it. ABS is also considered a food-grade plastic [42,43].

#### 2.2.3. Nylon

Nylon is a widely used engineering thermoplastic known for its excellent wear resistance and durability. The most commonly used grade of nylon for 3D printer filaments is PA 6. Nylon is both impact and wear-resistant, but it has a tendency to easily absorb moisture and requires relatively high print temperatures of up to 265 °C. Due to the high temperatures involved, nylon often warps during printing, making a heated enclosure recommended. Nylon expands when exposed to water because of its hygroscopic nature, and it can be dissolved by acetic acid and formic acid. There are food-safe grades of nylon available as well [44,45].

#### 2.2.4. Thermoplastic Polyurethane (TPU)

TPU is a flexible filament that is resistant to abrasion, grease, and oil. TPU boasts a Shore Hardness of 95 and provides a clean, easy printing experience. TPU is a strong yet bendable filament with excellent layer-to-layer bonding that prevents any layer separation. This flexible filament is semi-transparent and has a rubber-like appearance. Compared to other elastic filaments, TPU is one of the easiest to print, making it a favorite among novice makers [46].

#### 2.2.5. High-Impact Polystyrene (HIPS) 

HIPS is a thermoplastic commonly used for pre-production machining prototypes. It is also one of two 3D printing filament types used as a soluble support material, along with ABS. HIPS shares similar properties with ABS, making it an ideal second extruder material. However, it is important to note that HIPS emits harmful fumes during printing. Despite this, HIPS is known for its excellent durability thanks to its unique combination of flexibility and strength. One potential issue with HIPS is excessive warping if temperatures are not carefully controlled. It is worth mentioning that HIPS is soluble in D-limonene, and it is considered a food-safe material [47].

#### 2.2.6. Poly (Vinyl Alcohol) (PVA)

Polyvinyl alcohol is a biodegradable 3D printer plastic that dissolves easily in water. It also has similar printing properties to PLA, making PVA an ideal filament for PLA support material. However, PVA can be expensive because it is often used as sacrificial support material. Due to its water solubility, PVA is not suitable for most applications as moisture can degrade the plastic. PVA may warp slightly and will dissolve in water, so it is not recommended for use with food [48,49].

#### 2.2.7. Polyethylene Terephthalate Glycol-Modified (PETG)

PETG is a modified variant of Polyethylene terephthalate (PET). The addition of glycol lowers the melting temperature sufficiently for PETG to be more user-friendly. Aside from being easy to print, PETG is also UV-resistant. Its key disadvantages are its poor adhesion and its tendency to create strings when the printhead crosses empty space between features. PETG has excellent mechanical properties, while also being resistant to a wide range of chemicals and high temperatures. PETG is not particularly prone to warping. PETG is soluble in toluene and methyl ethyl ketone (MEK). PET is food safe and, by extension, so is PETG [50].

#### 2.2.8. Polycarbonate (PC)

PC is an advanced engineering thermoplastic known for its excellent mechanical properties, making it the strongest 3D printer filament available. With high strength and a glass transition temperature of 150 °C, it is ideal for applications requiring high temperatures. However, PC must be printed at temperatures as high as 310 °C. Due to its high hygroscopic nature, PC readily absorbs moisture, leading to potential defects in the printed parts. Despite being one of the most durable 3D printing filament options, PC is highly susceptible to warping. It can be dissolved in tetrachloromethane, pyridine, and chloroform. PC is commonly used for food containers [51,52].

## 3. FDM 3D Printed Fiber Reinforced Composites (FRCs)

A fiber-reinforced composite (FRC), as shown in Figure 3, is a building material composed of three components: reinforcing fibers (either continuous or discontinuous), a continuous phase matrix, and a fine interphase region, also known as the interface (where the different materials in the composite meet).

### 3.1. Composite Matrix

Regarding the matrix, two commonly used materials in reinforced composites as matrixes are thermoplastics and thermoset (thermosetting polymers). The former, thermoplastic, also known as thermosoft plastic, is a pliable plastic polymer that melts quickly at elevated temperatures and solidifies during the cooling process. The most widely used thermoplastics are PLA [53,54], ABS [37], polycarbonate (PC) [55,56], polypropylene (PP) [57], polyamide (PA) [58,59] (e.g., Nylon [60,61,62], polystyrene (PS), polyphenyl sulfone (PPSU), polyetheretherketone (PEEK) [63], polyaryletherketone (PAEK), and polyetherimide (PEI). Thermosets, conversely, are obtained by the irreversible hardening (curing) process of a soft solid or viscous liquid prepolymer (resin) [64]. Curing is induced by heat or suitable radiation and may be boosted by high pressure or the addition of a catalyst. Photo-curable resins, acrylic-based resins, and cyanate ether are among the commonly used thermosetting polymers [18,65,66].

### 3.2. Reinforcing Elements

#### 3.2.1. Fillers

Adding reinforcing fillers is one of the well-established methods to enhance the mechanical, electrical, and thermal properties of plastics. Using natural, mineral, or synthetic fillers not only reduces the price of end products, depending on the material used, but also positively affects mechanical and thermal properties, as well as thermal or electrical conductivity/resistivity, depending on the filler choice and target functionality. Fillers are classified based on their origin (natural or synthetic), chemical composition (organic or inorganic), as well as their shape, size, and aspect ratio. However, they are generally divided into: carbon materials (carbon black, graphene, nanotubes, carbon fibers), metallic and ceramic dusts, glassy and fibrous fillers (renewable raw materials such as hemp, kenaf, flax, jute, cellulose, bamboo, coconut, and others) mainly used to reinforce the structure and improve mechanical properties; mineral ones (titanium white, mica, metal powders, graphite, talc, chalk, diatomaceous earth), characterized by thermal, chemical, and UV resistance; and biofillers (coffee grounds, wood flour) [67].

#### 3.2.2. Fibers

As mentioned before, another part of an FRC is reinforcing composite, which is introduced to increase the composite strength. Fibers can be classified based on their length or their origin. Following is an introduction to different types of fibers.

##### Short Fibers

First, short fibers were used as reinforcing elements, leading to higher strength of printed parts. Being cost-effective and easy to use and having superior mechanical properties make short fibers attractive for reinforcing composites during 3D printing. Although adding short fibers can improve the mechanical characteristics of polymers, they can also make changes in polymers’ rheology, causing voids. One reason why short fibers are unable to significantly improve the strength is the reliance on the matrix material for transferring loads between fibers [28]. Another is the limited fraction volume (the percentage of fiber volume in the entire volume of fiber-reinforced composite material) of short fibers as it is often limited to a maximum of 50 percent to avoid the molten filaments’ high viscosity. Therefore, balancing the aspect ratio and volume fraction of fibers can help earn the optimum properties [25,28]. 

##### Continuous Fibers

In recent years, continuous fibers were introduced and developed to overcome the problems associated with short fibers [25]. Contrary to short fibers, continuous fibers can transfer and retain loads within unbroken strands of fibers, reducing the load applied to and transferred by the polymer matrix and leading to higher load-bearing capacity than composites made by short fibers. In composites with continuous fibers, the polymer matrix transfers off-axis loads among the fibers, like shear stresses. Therefore, composites fabricated with continuous fibers exhibit higher strength than those with short ones owing to their orientation [26].

In another classification, fibers can be divided into two groups: synthetic and natural fibers. Below are some frequently used fibers from both groups of natural and synthetic fibers.

#### 3.2.3. Synthetic Fibers

Synthetic fibers are fibers derived from raw materials such as petroleum and are based on chemicals or petrochemicals. These raw materials are polymerized into long, linear chemicals with various compounds and are used to produce different types of fibers. Synthetic fibers make up about 50% of all fiber used globally and have various applications in different industries. They are highly sought after for lightweight and innovative composite materials, playing a crucial role in the production of fiber-reinforced composites. Their use is growing worldwide due to their excellent properties and they are in high demand [68,69,70].

##### Carbon Fibers

Since carbon fibers are among the most widely used reinforcing elements, this section is specifically dedicated to discussing them. FDM of carbon fiber-reinforced polymers (CFRPs) combines the advantages of FDM, such as reduced prototyping and fabrication time, lower cost, customization, reduced emissions, and material waste, with the high strength of carbon fiber. As a result of these benefits, FDM of CFRPs has recently been applied in various fields, including aerospace, automotive, biomedical, and electronics [18,71,72]. Two types of carbon fibers are introduced in the next sub-sections.

Short carbon Fibers

Short carbon fibers, also known as discontinuous fibers, can be categorized into four groups based on their dimensions of length and diameter: nano, micro, milli-fibers, and nano-powders. Micro-fibers have a length ranging from 50 to 400 μm, while nano-fibers have a length of less than 1 μm. Milli-fibers, on the other hand, have a length in the millimeter range. For FDM 3D-printed composites, the best results are typically achieved using fibers with an aspect ratio (the ratio of length to diameter) of 1000 or more. Therefore, short carbon fibers with diameters between 5–7 μm and lengths between 5–7 mm lead to good mechanical properties [18,25,73].

The alignment, volume fraction, and length of the fibers have a significant impact on the mechanical properties of CFRPs made with short carbon fibers. During the FDM process, shear forces cause the reinforcing fibers to align in the print direction and remain aligned in the final printed part as the thermoplastic melts in the nozzle. This alignment results in anisotropic mechanical characteristics for CFRPs. Additionally, the volume fraction and length of the fibers affect the mechanical properties of the printed part. Studies [74,75,76] have shown that increasing the fiber content up to a certain point can effectively improve the mechanical properties. However, further increases in the volume fraction of the fibers negatively impact the strength of the printed part. This is due to increased porosity and interfacial debonding as the amount of reinforcing fibers increases [18].

2.Continuous carbon fibers

While short or discontinuous fibers play a positive role in improving the strength and elastic modulus of CFRPs, the ultimate strength and other mechanical properties are still lower than expected. This is because short fibers are the primary cause of fiber pull-out, which is a common failure mechanism in fiber-reinforced composite materials. Studies have also shown that longer carbon fibers, with their larger surface area, improve the adhesion between layers and result in improved mechanical properties. Therefore, continuous long carbon fibers are introduced to enhance the mechanical properties of the reinforced composites [77,78,79,80,81].

##### Glass Fibers

Glass fibers (GFs) are a highly versatile class of materials, commonly used as reinforcement fibers for polymeric materials. While the stiffness of glass fibers is lower than that of other reinforcement fibers, they offer the unique advantage of combining high strength with low density and a reasonable cost. Glass fibers are the least expensive option and are widely used in various industries such as electronics, aviation, civil engineering, and defense technology. They are valued for their excellent properties, including high strength, flexibility, stiffness, and resistance to chemical damage [82]. Various glass fibers with different mechanical properties are available for different applications. However, these fibers are primarily used as reinforcements for polymeric matrices and for in-vitro applications of bio-composites. The main advantage of glass fibers is their high performance-to-cost ratio [77,83,84].

##### Aramid

Aramid fiber was the first organic fiber used as reinforcement in advanced composites due to its high tensile modulus and strength. It possesses superior mechanical properties compared to steel and glass fibers of the same weight. Aramid fibers are naturally heat- and flame-resistant, allowing them to maintain their properties at high temperatures. However, they have poor resistance to ultraviolet light, causing fabrics made from aramid fibers to change color when exposed to it [85].

The most common type of aramid fiber is Kevlar, which was introduced later. The strength and modulus of aramid fibers are 5–6 times and 2–3 times higher than those of steel wires of the same diameter, respectively, while the fiber weight is just 1/5 of that of steel wire. Additionally, this fiber has excellent corrosion resistance properties and presents unique fatigue damage mechanisms compared to other commonly used reinforcement fibers in composites. Aramid fibers are commonly used in bulletproof vests, blast protection systems, cooling instruments, ship hulls, and micro-strip antennas for spacecraft [86,87,88,89].

##### Kevlar

Kevlar is a synthetic fiber that possesses high strength, durability, toughness, thermal stability, and elastic modulus. It belongs to the aromatic polyamide family, which means it is made from a class of synthetic polymers called aromatic polyamides. Aramid fibers, like Kevlar, can be used as a great alternative to carbon or glass fibers or in conjunction with them. Kevlar-reinforced composites find applications in various products, including boats, airplanes, automobiles, sporting goods, and consumer products. One of Kevlar fiber’s main advantages is its ability to withstand high temperatures and resist abrasion. It makes it a popular choice for products that operate under extreme conditions [90,91]. Table 1 summarizes the different composites with different matrix materials and synthetic fibers and how their properties have been changed. Based on Table 1, Carbon, Glass, and Kevlar fibers are the most widely used fibers for reinforcing polymeric materials. It is also clear from Table 1 that adding synthetic fibers can significantly improve the mechanical properties of the matrix.

#### 3.2.4. Natural Fibers

Due to growing environmental concerns, the development of polymer composites using materials that can be decomposed or recycled is crucial [101]. Replacing synthetic and carbon fibers with natural fibers offers numerous advantages that mitigate the negative effects of synthetics, such as air pollution, respiratory problems, recyclability, sustainability, mechanical characteristics, and waste issues [102,103].

Different countries around the world cultivate and utilize a variety of natural fibers, often engaging in import and export activities with other regions. For instance, European automotive industries predominantly utilize flax and hemp in their products. Additionally, they import other fibers like kenaf and jute from different countries. Figure 4 illustrates the fibers utilized by European automotive industries in 2012 [104].

##### Flax

Flax is one of the strongest natural cellulosic fibers. It was the first plant stem fiber used for making textiles. Flax fiber is extracted from the skin of the flax plant’s stem. It is a soft, lustrous, and flexible fiber that is stronger than cotton but less elastic. It can be used as a reinforcing material in composite materials and has the potential to be used as load-bearing constituents in composites due to its attractive properties, such as a high stiffness-to-weight ratio. Flax is also used in the food production industry, personal care products, animal feeds, and various industrial applications [105,106].

##### Cotton

Cotton fibers are a group of natural hollow fibers known for being breathable and absorbent. They are the purest form of cellulose, which is the world’s most abundant natural polymer. Approximately 90% of cotton is composed of cellulose, making it the most widely used fiber in the textile industry. Cotton fibers can hold water 24–27 times their own weight, making them excellent at absorbing moisture. They are also strong, absorbent of dyes, and can withstand abrasion wear and high temperatures. In short, cotton is comfortable. However, cotton fibers are prone to creasing and shrinkage. Creasing can affect the aesthetic look of a product, while shrinkage can lead to dimensional changes. To address these issues, cotton fibers can be mixed or blended with other materials like polyester or nylon or treated with a permanent finish to improve their properties and overcome their shortcomings, enhancing their end-use characteristics [107,108,109,110].

##### Kenaf

Kenaf fiber is a well-known natural fiber used to reinforce polymer matrix composites, preferably sourced from the bast part of the kenaf plant. It has the potential to replace synthetic fibers like glass fiber, providing mechanical properties such as tensile strength comparable to synthetics but with lower density. This results in lightweight and environmentally friendly polymer composites. Using kenaf fiber as reinforcement can also decrease the wear rate of polymer composites. For example, regardless of fiber orientation, studies have shown that kenaf fibers enhance the tribological properties of epoxy. Interestingly, the wear rate of the epoxy decreased even more when the fiber orientation was perpendicular to the sliding direction [111,112].

##### Hemp

Hemp fibers are some of the strongest members of the bast natural fibers family, derived from the hemp plant within the Cannabis species. Nowadays, these fibers are gaining wider applications as they are used as reinforcements in composite materials due to their biodegradability and low density compared to artificial fibers. Additionally, these materials possess inherent mechanical, thermal, and acoustic properties. Surface functionalization of hemp fibers is of significant importance to expand their applications [113].

##### Wood

Wood fibers are typically cellulosic elements extracted from trees and used to create materials like paper. When combined with thermoplastics, wood fibers can produce durable, waterproof products suitable for outdoor use, such as deck boards or outdoor furniture. Wood fiber has a high total porosity, with a typically high level of air-filled porosity and a low level of readily available water [114]. Wood-plastic composites (WPCs) have been utilized in various molded applications typically seen with standard thermoplastics. The enhanced mechanical, thermal, and processing properties of these materials have enabled their widespread use in intricately shaped parts within the automotive and building products sectors [114].

##### Jute

Jute fiber is produced from plants in the genus Corchorus, in the Malvaceous family. Jute is a lignocellulosic fiber that is both a textile fiber and a type of wood. It falls into the bast fiber category, which refers to fibers collected from the bast or skin of the plant. Jute fibers are completely biodegradable and recyclable, making them environmentally friendly materials. They have good insulating properties for both thermal and acoustic energies, along with moderate moisture regain and no skin irritations [109,115].

##### Basalt

Basalt fibers are created by melting crushed basalt rocks at 1400 °C and then drawing the molten material. These fibers possess superior mechanical and physical properties compared to glass fibers. Their main advantages include fire resistance, good resistance to chemically active environments, as well as vibration and acoustic insulation capabilities. Improved production facilities and quality control capabilities allow for the fabrication of high-quality basalt fibers with minimal variability in properties. While basalt fibers are more costly than E-glass fibers, they are significantly cheaper than carbon fibers [116]. Table 2 summarizes the different composites with different matrix materials and natural fibers and how their properties have been changed.

## 4. FDM of Continuous Fibers

Adding continuous fibers to polymers and creating Fiber-Reinforced Composites (FRCs) improves the mechanical behavior of polymers. This includes an increase in strength and stiffness, thermal conductivity, and a reduction in thermal expansion and warpage, when compared to other reinforcements. However, there is still a difference between the mechanical properties of FRCs made with the FDM process and those of composites produced via conventional fabrication methods. Therefore, it is necessary to explore the mechanisms of the FDM process in order to find new ways to improve the mechanical properties of reinforced components fabricated with FDM. The goal is to make them at least comparable to those of composites made through conventional processes. Different FDM mechanisms of Continuous Fiber-Reinforced Composites are provided in the following sub-sections.

### 4.1. In-Situ Fusion Mechanism

This system utilizes two input materials: the reinforcement (dry fiber feedstock or reinforcing fiber) and the neat/pure polymer matrix. Both the reinforcement and matrix are combined during the printing process in this system. One commonly used technique in this mechanism is known as “nozzle impregnation”. During this process, the reinforcing filament/dry fiber feedstock is drawn into the nozzle and preheated, while, the matrix polymer is introduced into the melt zone via a motor-driven hobbed gear at the same time. In the melt zone, the melted polymer and the preheated filament come together due to the pressure of the polymer being fed into the melt zone via the motor-driven hobbed gear, and they are finally deposited together.

As mentioned earlier, in this mechanism, both fibers and thermoplastics are simultaneously drawn into the nozzle. Consequently, the user has control over the flow rate of the thermoplastic content and can adjust it according to the application of the printed part. Another advantage of this mechanism is that it is a single-step manufacturing method.

However, a drawback of this method is the poor bonding between the layers (fiber and matrix) due to the short dwell time, which results in a weakened strength of the printed part. The quick dwell time in the nozzle leads to inadequate polymer infusion into the fiber bundles, resulting in increased porosity and weaker mechanical properties. Consequently, this mechanism produces a subpar fiber-matrix interface due to low compaction during the printing process [18,28].

Nakagawa et al. [35] employed this method to produce carbon fiber-reinforced plastic components. They achieved this by inserting bundled carbon fibers (the reinforcing fibers) into the plastic ABS filament (the matrix material) through a single nozzle and extruding them simultaneously. The bundled carbon fibers adhered to the matrix material filaments by undergoing heating while passing through a single nozzle with an entry diameter of 2.5 mm and exit diameters of 0.4 mm and 0.9 mm [35]. The extruded ABS filament with a diameter of 1.75 mm and a tensile strength of 30 MPa was used. The carbon fibers, measuring 6 μm in diameter and possessing a tensile strength of 5.3 GPa, were embedded between ABS filaments. They reported three results for both exit diameters of the single nozzle, for pure ABS with reinforcing fibers, for the ABS-reinforced printed parts with carbon fibers sandwiched within the ABS filament with and without thermal bonding (with heating pin).

According to their results, sandwiching fibers alone did not increase the nominal tensile strength. However, thermal bonding significantly improved the tensile strength of both composites extruded through the nozzle with 0.4 and 0.9 exit diameters. They also showed that samples with a nozzle diameter of 0.9 mm had some cavities, resulting in lower tensile strength compared to those deposited from the 0.4 mm diameter nozzle, which bonded the fibers appropriately and sufficiently. The researchers also compared the results of using a microwave oven for thermal bonding with those obtained using a heating pin. The results showed no significant changes in the tensile strength of samples where ABS and carbon fibers were bonded using either microwave or heating pin thermal bonding. Additionally, thermal bonding increased the bending load compared to samples without thermal bonding (using microwave and/or heating pin) [35].

Matsuzaki et al. [36] also utilized the same method of in-nozzle impregnation to produce Filled and Reinforced Thermoplastics (FRTP). In their study, they separately fed the thermoplastic resin (polylactic acid (PLA) resin) and reinforcements (carbon fiber tow (CFRTP) and twisted jute fiber yarn (JFRTP)) into the printer head. The reinforcing fibers were preheated using a nichrome wire before entering the nozzle, while the matrix material was melted by a heater within the printer head. The resulting reinforced composite was then deposited layer-by-layer on the hot build platform (80 °C) using a single nozzle. Within the nozzle, the matrix material filament and both reinforcing fibers were heated to 210 °C. The feeding rate and fraction volume for CFRTP were 100 mm/s and 6.6%, respectively, while for JFRTP, they were 60 mm/s and 6.1%. Figure 5 provides a comparison of the tensile strength between pure FRTP, CFRTP, and JFRTP [36].

According to this figure, reinforcing with carbon fiber tow resulted in a much higher tensile strength than unreinforced PLA and samples reinforced with twisted jute-fiber yarn. For CFRTPs samples, the tensile stress increased linearly up to the maximum value, with a negligible drop in stress before fracture. However, for the other two samples, unreinforced PLA and JFRTPs, the stress-strain curves were not linear, and no drop in stress before fracture was reported. The tensile strength of JFRTPs samples was only slightly higher than that of unreinforced PLA. Both CFRTPs and JFRTPs samples exhibited fiber pull-out, which is a common weak point in reinforced composites. This indicates poor adhesion between the reinforcing fibers and the matrix material (see Figure 6).

Yang et al. [37] utilized a similar approach to create Continuous fiber-reinforced thermoplastic composites (CFRTPCs). In their modified method, they introduced a new extrusion head that could receive both the reinforcing fibers and matrix filament and heat them to a semiliquid state within the nozzle. The reinforcing fibers were drawn from a fiber supply coil and passed through the extruder’s inner core, allowing for infiltration and coating with molten matrix material. Ultimately, the reinforced composite was extruded from the nozzle. ABS with a 1.75 mm diameter was used as the matrix material, while carbon fiber (1000 fibers in a bundle) was used as the reinforcement with a volume fraction of 10 wt. %.

To evaluate the performance of their CFRTPCs, the researchers conducted a three-point bending test, as well as tensile and interlaminar shear tests. They compared the results with those obtained from 3D printed unreinforced ABS fabricated using FDM (ABS reinforced composite fabricated via injection molding with 10% carbon fiber). The findings showed that reinforcing ABS with carbon fibers through the in-situ mechanism of the FDM process significantly increased the flexural strength from 80 MPa for FDM 3D-printed unreinforced ABS to 127 MPa, which was close to the strength of the reinforced ABS with continuous carbon-fiber (CCF/ABS) fabricated using the injection molding process (140 MPa).

In terms of tensile strength, although reinforcing ABS through the FDM process greatly improved the tensile strength of unreinforced 3D printed FDM ABS from 50 MPa to 147 MPa, the tensile strength of the CFRTPC was still lower than that of CCF/ABS with injection molding (200 MPa) [37].

In both cases, the fracture mode was surface tension fracture. Initially, the matrix material covering the reinforcing fibers fractured due to tension, followed by the pulling out of the reinforcing fibers from the matrix. Fiber breakage persisted until the fibers inside the matrix ruptured.

### 4.2. Dual Extruder Mechanism/Ex-Situ Method

In this method, the FDM process is based on fabricating the part using two extruders: one for heating and depositing the pure polymer filament, and the other for depositing the reinforcing filament. The reinforcing filament, known as prepreg filaments, is fabricated before printing by combining and mixing fibers with thermoplastics/polymers.

MarkForged is one of the most well-known companies in the world that produces continuous fiber-reinforced prepregs filament for CFRC 3DP. The polyamide resin used by MarkForged is suitable for impregnating continuous carbon fiber, glass fiber, aramid fiber, and other materials to create CFRPF/PA products. Anisoprint is another company that offers continuous carbon fiber-reinforced composite materials (CCFRC) and continuous basalt fiber composite materials (CBFRC). The continuous fibers used by Anisoprint are pre-impregnated and solidified in thermosetting resin materials [124].

The prepreg filament is then extruded simultaneously with the polymer filament through dual extrusion. By using this FDM mechanism, the user can reinforce specific layers and have control over the content and position of the reinforcing filament in those layers.

This mechanism also has its own advantages and disadvantages. While adding one more extruder leads to an increase in the printer cost and maintenance, uses more filaments, and can be time-consuming due to balancing parameters for each extruder, there are some advantages to this mechanism, such as strengthened infill, greater flexibility, and precision. This mechanism allows printing two identical parts simultaneously, enhancing mechanical properties. Another big advantage of independent dual extruders is that it allows the two nozzles to have a big temperature difference, therefore enabling the combination of different materials. Since the heads are separated from each other, it is possible to set the nozzles at different temperatures with a large difference between them. This allows the user to create many more material combinations.

The dual extruders system allows printing two identical parts simultaneously, doubling the production speed of the printer and making the machine much more effective for serial manufacturing.

Continuous carbon fiber reinforced polyamide composite (CCF/PA) parts were fabricated using the FDM mechanism, based on a patented dual extrusion FFF technology, via the Mark Two^®^ 3D printer from MarkForged^®^ (Watertown, MA, USA) by Lupone et al. [125].

The Markforged Two is a dual extruder printer equipped with two extrusion nozzles that allow the printing of two spools of filaments, one of a plastic matrix and one of reinforcing continuous fibers, respectively (Figure 7a). The CCF/PA samples were constructed using four different layups: longitudinal (referred to as (0)), cross-ply (referred to as (0,90)_s_), quasi-isotropic (0/±60)_s_, and (0/+45/90/−45)_s_, The aim of the study was to investigate the mechanical and microstructural properties of 3D printed CCF/PA composites with various layups.

The typical stress-strain curves of CCF/PA samples with different layups are shown in Figure 8. The composites exhibited linear elastic behavior until failure, indicating that the fibers effectively withstand most of the applied stresses. A very low strain at break (in the range of 1–1.2%) was recorded, which is typical of brittle materials (Figure 8).

Melekna et al. [6] conducted a study on the elastic properties of 3D-printed structures using a nylon filament and varying volume fractions (4.04%, 8.08%, and 10.1%) of continuous Kevlar fibers. The choice of Kevlar was based on the research group’s prior experience with the material. The researchers utilized the Volume Average Stiffness method (VAS) to predict the resulting properties.

The print parameters used in the study were as follows: each layer had a height of 0.1 mm, the infill percentage was 10%, and the orientation was set at 45 degrees. The shell thickness was 0.4 mm, with two shell layers. The number of infill, floor, ceiling, and solid layers were 8, 4, 4, and 8, respectively. To reinforce the 3D-printed test specimens, concentric fiber rings (Kevlar rings) were employed. These test specimens were prepared in accordance with ASTM D638-14. One sample was printed without any rings, while three others were printed with two rings (4.04% volume fraction), four rings (8.08% volume fraction), and five rings (10.1% volume fraction), with the neck region limited to 13 mm. The aim was to assess the impact of these reinforcements on the properties of the samples, as depicted in Figure 9a,c.

Four different regions can be observed in Figure 9b: two gray regions represent shell layers that surround the external area of the structure, with nylon filament deposited along the longitudinal axis of the sample. Additionally, there are white solid layers consisting of compact layers of nylon, with orientations alternating between ±45 degrees from the longitudinal axis. The yellow regions correspond to Kevlar layers with longitudinally oriented concentric rings. Infill regions can also be observed in the middle of the Kevlar layers.

To investigate the internal structure and failure mechanism of the printed samples, an optical microscope was utilized in their study. Additionally, Malekna et al. developed a mathematical/analytical predictive model based on the VAS method to determine the effective elastic constants of the printed composite. This model considers the elastic constants of each region, as each part of the printed composite possesses its own elastic constant. Consequently, the model predicts the effective constant by taking into account the elastic constants of all regions, thereby contributing to the overall constant of the entire composite.

Figure 10 displays the elastic modulus obtained from experimental tests compared to the values obtained from the analytical model.

Based on Figure 10, there was an acceptable agreement between the experimental and analytical model results. It is also clear that by using reinforcing rings and increasing the number of rings, the tensile properties also increased.

Additionally, the researchers observed that all sample fractures occurred at the location where the fiber started to deposit (see Figure 11). Therefore, the starting point of the fiber is of great importance, as it has lower strength compared to other parts. Consequently, the 3D printed part is more likely to fail in this region.

Van Der Klift et al. [38] also utilized a similar technique to create 3D-printed continuous carbon fiber reinforced thermoplastic (CFRTP) using FDM and examine its tensile properties. They employed Nylon FFFR as the matrix material positioned in the center of a square-shaped sample measuring 45 mm × 45 mm × 3 mm. Carbon fibers (CFRTP layers) were then applied continuously along the edges of the square, as shown in Figure 12. However, the figure reveals that there are visible gaps in the reinforcement deposit, indicating that the carbon fibers were not continuous in certain areas. This lack of continuity can be attributed to the printing pattern, which aims to prevent the printer head from becoming stuck as it approaches the surface of the sample.

They prepared three samples: the first one with ten layers of pure Nylon, the second (6CF) with two layers of Nylon on each side plus six layers of CFRTP in the center, and the third one (2CF) with four layers of Nylon on each side plus two layers of CFRTP in the center (Figure 12c–e). By conducting tensile tests on three types of samples (specimens with and without tapered tabs, 6CF and 2CF, and pure nylon), they observed that for pure Nylon, the average tensile strength was about 17 MPa, while for 6CF, it was in the range of 370 Mpa and 520 Mpa, and for 2CF, the value was between 128 Mpa and 171 Mpa. The result showed that 6CF samples had higher tensile strength than 2CF, and samples fabricated with pure Nylon had the lowest amount.

According to their results, for 6CF samples, failure occurred in the vicinity of tabs, where the samples are clamped, while this place was not the one with the smallest cross-section.

As mentioned earlier, continuous fibers are developed to substitute short fibers and increase structural composites’ performance in terms of their mechanical properties. However, for these composites to be used as engineering materials and in engineering applications, it is essential to evaluate the influence of environmental conditions on their different mechanical properties. One of these environmental conditions is moisture, especially for polymers like PA, which are moisture sensitive. Therefore, Ghabaud et al. [25] used the same mechanism to fabricate composites with PA 6 as the matrix material and continuous carbon or glass fibers coated with the PA matrix and studied the effect of moisture on the properties of the resultant composite. Pure PA 6 was also printed as the top and bottom layer for each sample, preventing the part from dismantling. They prepared three types of samples with different fraction volumes and thicknesses.

They first kept tensile test samples in enclosures with Relative Humidity (RH), which was regulated by a saturated solution of Potassium-Hydroxide (KOH), Magnesium-Chloride (MgCl_2_), Sodium-Chloride (NaCl), and Potassium-Sulfate (K_2_SO_4_) to obtain an RH of 10%, 30%, 75%, and 97%, respectively. The samples were then stored at 50% humidity in the test room at 23 °C. They also studied the porosity and microstructure of the composite using ASTM D2734-16 2009 and an optical microscope, respectively. They observed that 9–98% moisture content could decrease the longitudinal tensile modulus and tensile strength of samples reinforced with continuous carbon fibers and the thickness of 1mm by 25% and 18%, respectively. Moreover, the debonding between the layers is more significant at 95% RH than at 15% RH. For samples with continuous glass fibers, the tensile modulus was stable in different moisture contents, but the tensile strength decreased by 25% with increasing the RH. Therefore, water sorption can significantly weaken the mechanical properties of the composite and degrade the matrix/fiber interface adhesion. Samples with carbon fibers also showed 40% more internal porosity than those with glass fibers, which is attributed to the higher porosity of carbon-based filaments than the glass-based ones [25].

Caminero et al. [126] also studied the impact damage resistance of FDMed composites reinforced with continuous fibers using the same FDM mechanism, where two nozzles are present; one prints the matrix material, and the other prints reinforcements. Nylon filaments with a diameter of 1.75 mm were used as the matrix material, and three continuous fibers made out of carbon, glass, and Kevlar with bundles of 0.35 mm, 0.3 mm, and 0.3 mm, respectively, were used as reinforcements. Three energy absorption and dissipating mechanisms, including fiber pull-out, delamination, and fiber breakage, define the impact resistance properties of reinforced composites under loading, leading to weakened load-carrying capacity. Therefore, studying the interfacial characteristics of the composite as well as fiber content is of great importance [126].

The Charpy test was long used for metals for low-velocity impact circumstances and has now been extended for polymers. Therefore, Caminero et al. used this test to investigate the impact damage performance of the fabricated reinforced composite. They used two build directions, flat and on-edge, where both matrix and reinforcement fibers are oriented and deposited along the longitudinal direction. Two types of specimens were also prepared, one with pure Nylon and the other with reinforcements, Figure 13a,b. While pure nylon samples were prepared in three thicknesses of 0.1, 0.125, and 0.2 mm, reinforced samples were prepared in three types (Type A, Type B, and Type C) with different fraction volumes for each reinforcement (Figure 13d).

SEM images of three types of fibers plus the interface between the glass fiber bundles and matrix (nylon) are also shown in Figure 13c.

For unreinforced samples (samples fabricated with pure Nylon), build orientation significantly affected the impact strength, so Flat samples showed higher impact strength than One-edge ones in all thicknesses. For Flat samples, increasing the thickness from 0.1 mm to 0.125 mm and, lastly, 0.2 mm dramatically increased impact strength from 20 MPa to 40 MPa. Unlike Flat samples, for which there was a direct relationship between the thickness and the impact strength, for One-edge samples increasing thickness led to a reduction in the impact strength from 20 MPa to just below 10 MPa.

One-edge samples had slightly higher impact strength for reinforced samples with carbon, glass, and Kevlar fibers than Flat ones. Increasing the volume fractions (changing from Type A to C) increased impact strength in both building orientations. Generally, samples reinforced with glass fibers had the highest impact strength compared to carbon and Kevlar fibers, with maximum impact strength between 250–300 MPa. However, samples fabricated via carbon fibers had the weakest impact resistance, showing a maximum of 50 to 80 MPa impact strength for One-edge and Flat samples. Samples reinforced with Kevlar fibers had a maximum impact strength of 80–200 MPa for One-edge and Flat samples. Moreover, according to Figure 14, samples with carbon fiber showed more brittle facture than two other fibers. Table 3 summarizes the studies conducted on this FDM mechanism.

### 4.3. Other FDM Mechanisms

#### 4.3.1. 3D Compaction Printing

Ueda et al. [129] developed a 3D compaction printing mechanism (3DCP) by attaching a hot compaction roller to increase the adhesion between deposited layers (beads) to lessen the voids and promote the mechanical behavior. Figure 15 shows the schematic of this modified FDM printing equipment with a compaction roller made from aluminum and with a 10 mm diameter. In this system, the compaction term refers to pressing filament against the printer bed after it is injected from the guiding nozzle [129]. In other words, the roller’s main role is pressing the printed filament on the previous layers immediately after it is deposited from the nozzle to reduce the voids and porosities and increase the adhesion between printed layers. According to Figure 15, in addition to the roller, a cartridge is also attached to the nozzle performing as a fixed shaft for supporting the roller. The roller has some internal bearings for making the roller able to rotate. The roller is heated via the attached cartridge heater. In their study, Ueda et al. [129] prepared CFRTP via conventional 3D printers and 3D compaction printers to compare their results. Based on their results, 3DCP could increase the tensile strength of CFRTPs by 33% compared to that of samples printed using conventional 3D printing. However, there was no significant difference between the modulus of both samples. Bending test results also showed 26% and 62% improvement in flexural modulus and flexural strength using 3DCP compared to 3DP. XCT and SEM images also exhibited many large voids in specimens fabricated by conventional 3D printers in contrast to dispersed small voids in samples with 3D compaction printers.

#### 4.3.2. Modified In-Situ Fusion Mechanism

Akhoundi et al. [130] developed a modified FDM printer by designing a nozzle into which continuous glass fiber was drawn and guided exactly at the melt zone and through an orifice plate attached to the side of the nozzle and then impregnated by the molten matrix material filament and immediately extruded on the build platform or previous layers (Figure 16).

As shown in Figure 16, there is a fixed pulley used for fixing the continuous glass fiber yarn and an idle pulley de-coiling the fiber yarn and feeding it into the molten region as the nozzle moves. They used PLA as a matrix material in 1.75 mm diameter and continuous glass fiber yarn as the reinforcing element with different fraction volumes. They compared tensile strength and modulus obtained from experimental and theoretical analysis through the mixture rule. Their results showed a good agreement between data obtained from experimental tests and theoretical analysis, which indicates a sound glass fiber depositing. Changing the faction volume from 49.3% to 35.1% decreased tensile modulus and strength. Based on their claims, the main advantage of their method was the system’s capability for online changing of the fiber fraction volume.

Table 4 compares the introduced FDM mechanisms.

## 5. Conclusions

Continuous and discontinuous/short reinforcing fibers have been introduced to enhance the mechanical properties of 3D-printed FDM parts, resulting in fiber-reinforced composites. Continuous fibers have shown significant potential in improving the mechanical properties of these printed composites, particularly in terms of their load-bearing capacity. As a result, researchers have predominantly focused on continuous fibers rather than short fibers. There are different mechanisms available for manufacturing reinforced composites using FDM technology and continuous fibers. Two main mechanisms are in-situ fusion and ex-situ prepreg. The in-situ fusion method involves drawing two input materials into a single nozzle, heating them, and combining them before printing. This method offers the advantage of allowing the operator to control the fiber content, and it is also a cost-effective process that can be completed in a single step. However, one drawback of this method is the insufficient adhesion between the deposited layers. On the other hand, the ex-situ prepreg mechanism involves using two extruders, one for depositing the reinforcing fiber and the other for the matrix material. This method is more expensive and time-consuming compared to the in-situ fusion mechanism. Both mechanisms have been found to result in the presence of voids and porosities in the fabricated parts. To address these issues, researchers have developed new mechanisms by modifying the two main mechanisms mentioned above. For example, compaction rollers can be attached to the FDM machine to apply more pressure on the printed layers, thereby filling the voids and gaps and leading to improved mechanical properties.

## Figures and Tables

**Figure 1 polymers-16-00831-f001:**
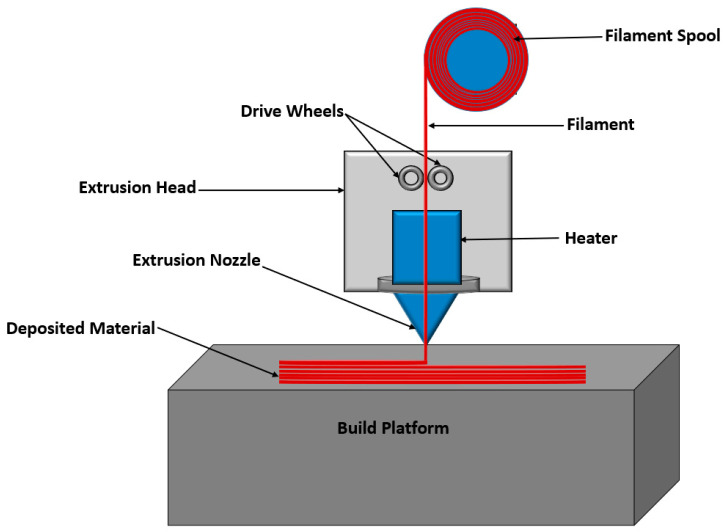
Schematic of a simple FDM process.

**Figure 2 polymers-16-00831-f002:**
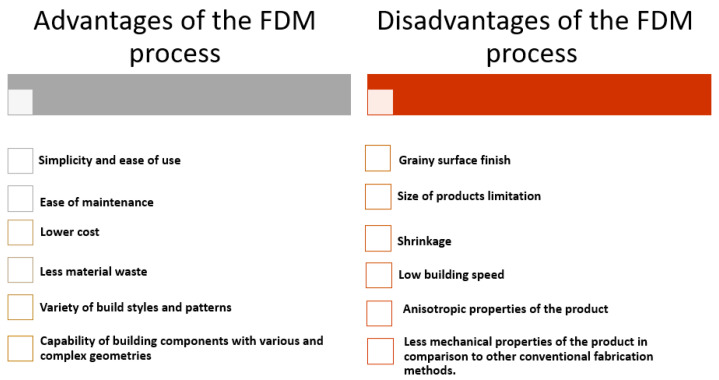
Advantages and disadvantages of the FDM process.

**Figure 3 polymers-16-00831-f003:**
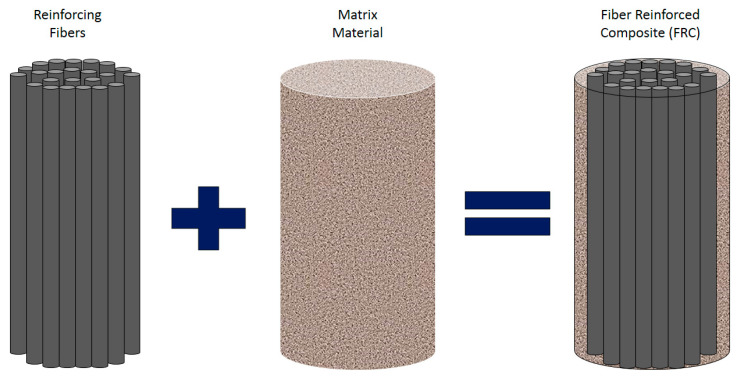
Schematic of a reinforced composite with reinforcing fibers.

**Figure 4 polymers-16-00831-f004:**
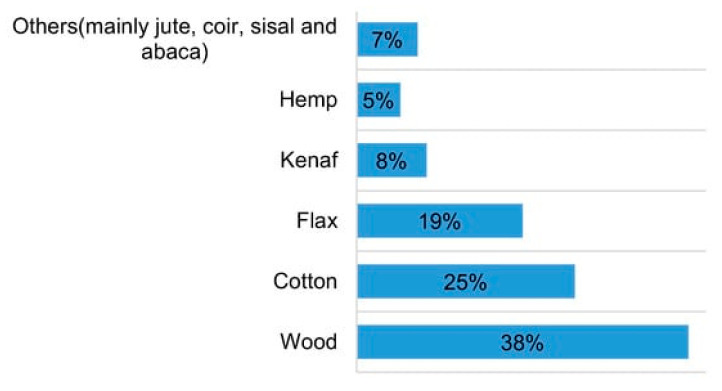
Utilization of wood and other natural fibers in automotive industries in Europe in 2012 [104].

**Figure 5 polymers-16-00831-f005:**
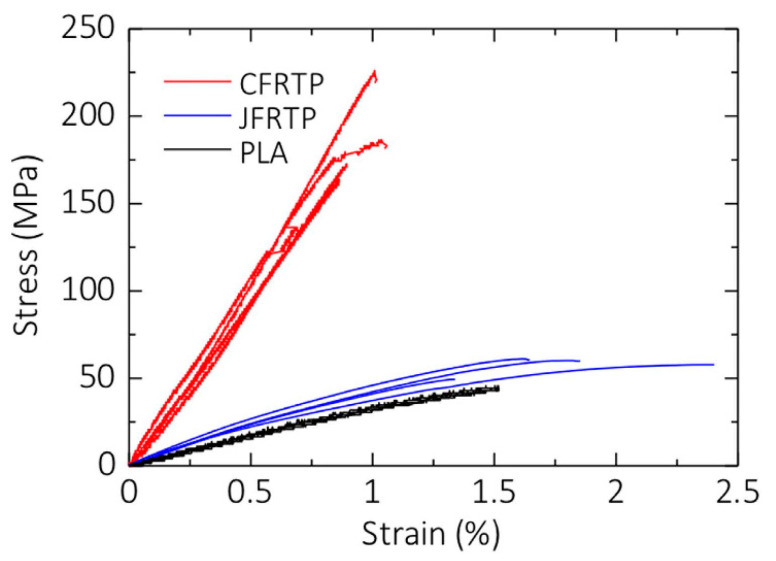
Stress-strain curves of PLA, CFRTP and JFRTP samples [36].

**Figure 6 polymers-16-00831-f006:**
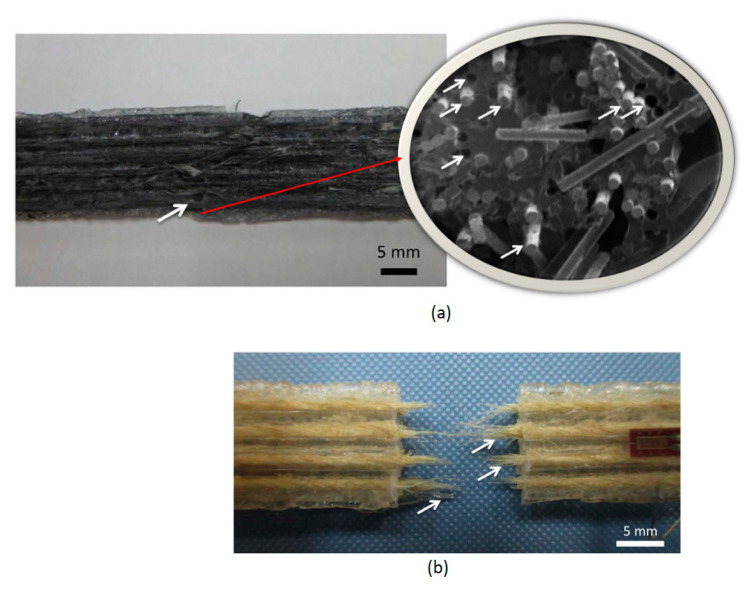
(**a**) Macroscopic and microscopic images of fractured CFRTPs and (**b**) macroscopic image of JFRTPs after fracture [36].

**Figure 7 polymers-16-00831-f007:**
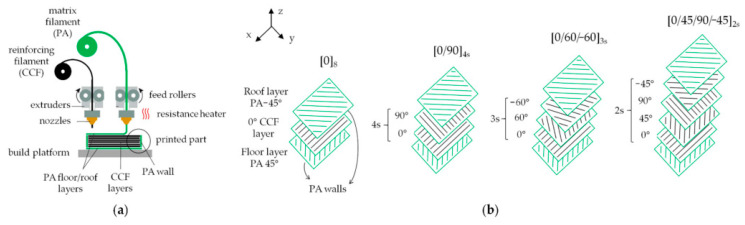
(**a**) Schematic representation of the MarkForged FFF printing process; (**b**) internal structure of the CCF/PA composites includes PA roof/top layers, CCF reinforced intermediate layers displaying fibers infill with different orientations (0°, 90°, 45°, 60°) based on the layup used, and a PA contour for each layer [125].

**Figure 8 polymers-16-00831-f008:**
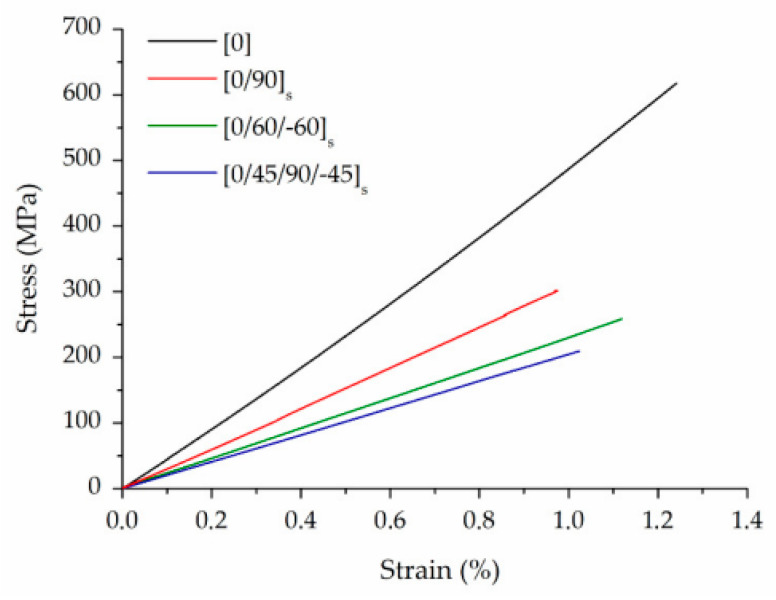
Typical stress-strain curves of CCF/PA composites with different layups [125].

**Figure 9 polymers-16-00831-f009:**
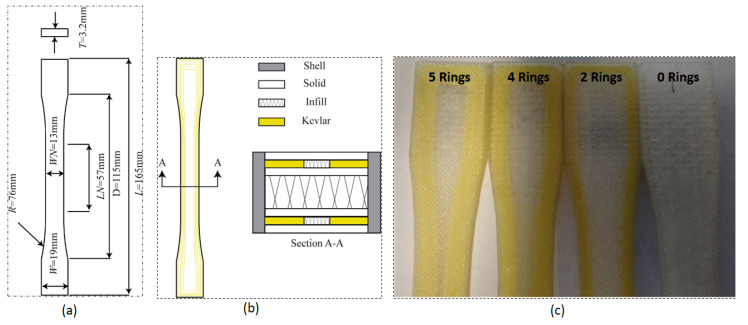
(**a**) Schematic of the ASTM standard used for tensile test specimens, (**b**) cross-section view of the printed parts, and (**c**) 3D-printed tensile samples with nylon filament and Kevlar rings [6].

**Figure 10 polymers-16-00831-f010:**
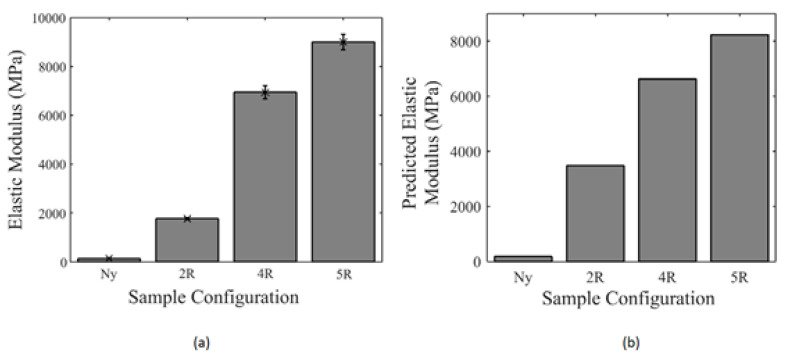
A comparison between the elastic modulus of specimens obtained from (**a**) experimental tests and (**b**) analytical model [6].

**Figure 11 polymers-16-00831-f011:**
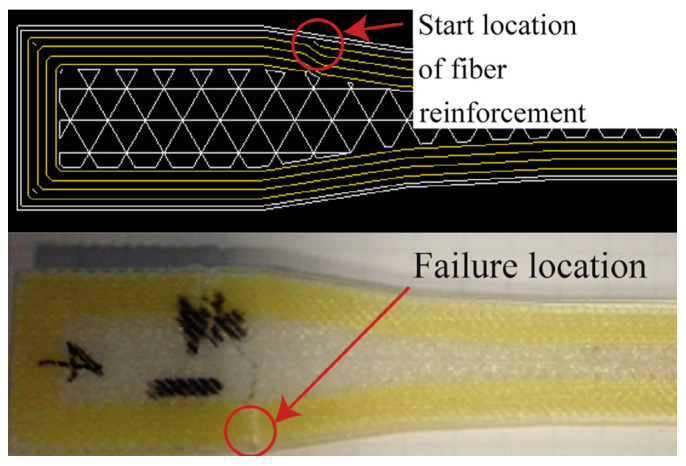
Failure location of a sample with four rings [6].

**Figure 12 polymers-16-00831-f012:**
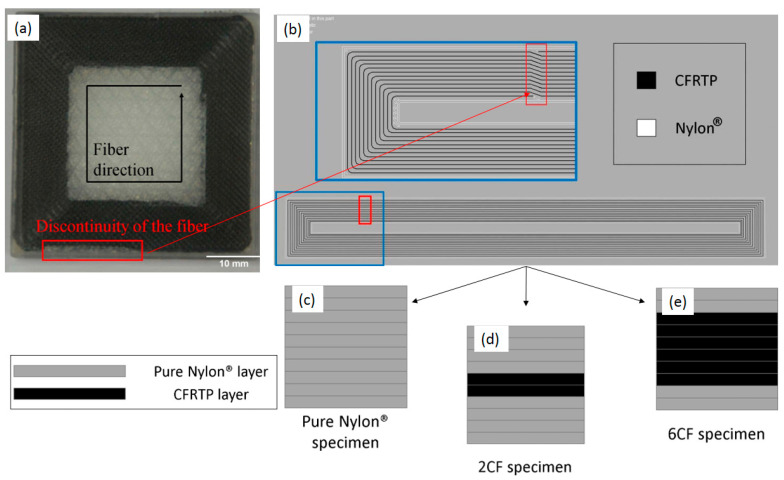
(**a**) 3D printed FDM composite with matrix material in the center and reinforcing fibers surrounding it and the discontinuity of the fibers (**b**) schematic of the printed reinforced composite showing the discontinuity of the reinforcing fibers around the matrix material, and schematic of three different 3D printed FDM parts with (**c**) pure Nylon, (**d**) two layers of reinforcing fibers, and (**e**) six layers of reinforcing fibers [38].

**Figure 13 polymers-16-00831-f013:**
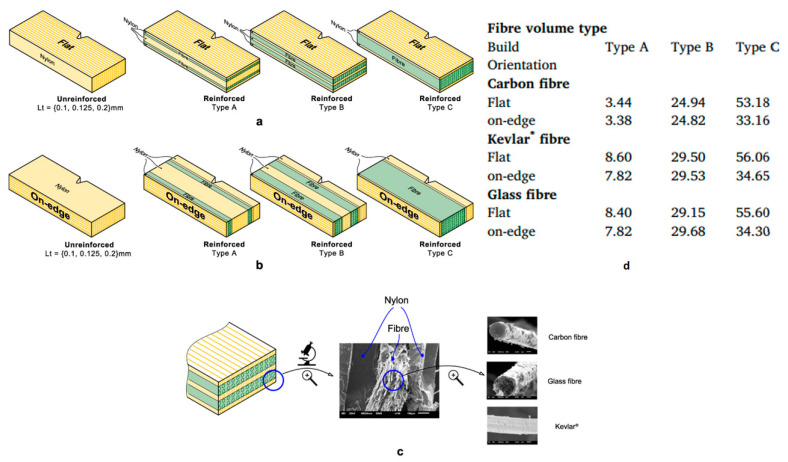
(**a**) Flat samples, unreinforced, and Type A-C, (**b**) One-edge samples unreinforced, Type A-C, (**c**) SEM images of a reinforced composite and the fibers, and (**d**) fraction volumes of three reinforcing fibers in three Types of A, B, and C for Flat and On-edge samples [126].

**Figure 14 polymers-16-00831-f014:**
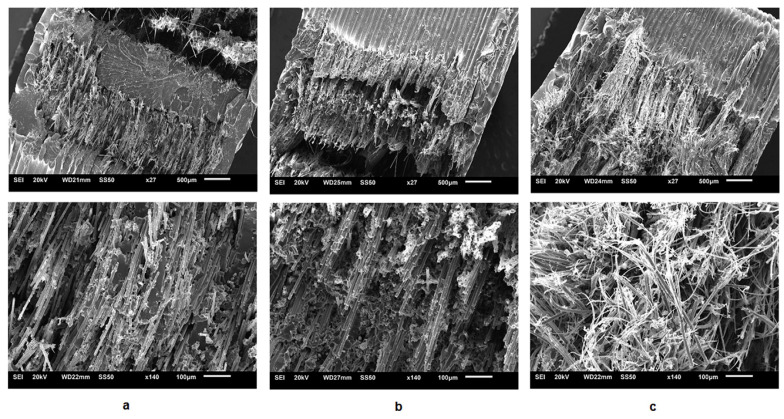
SEM images of Flat samples in Type C with (**a**) carbon fiber, (**b**) glass fiber, and (**c**) Kevlar fiber [126].

**Figure 15 polymers-16-00831-f015:**
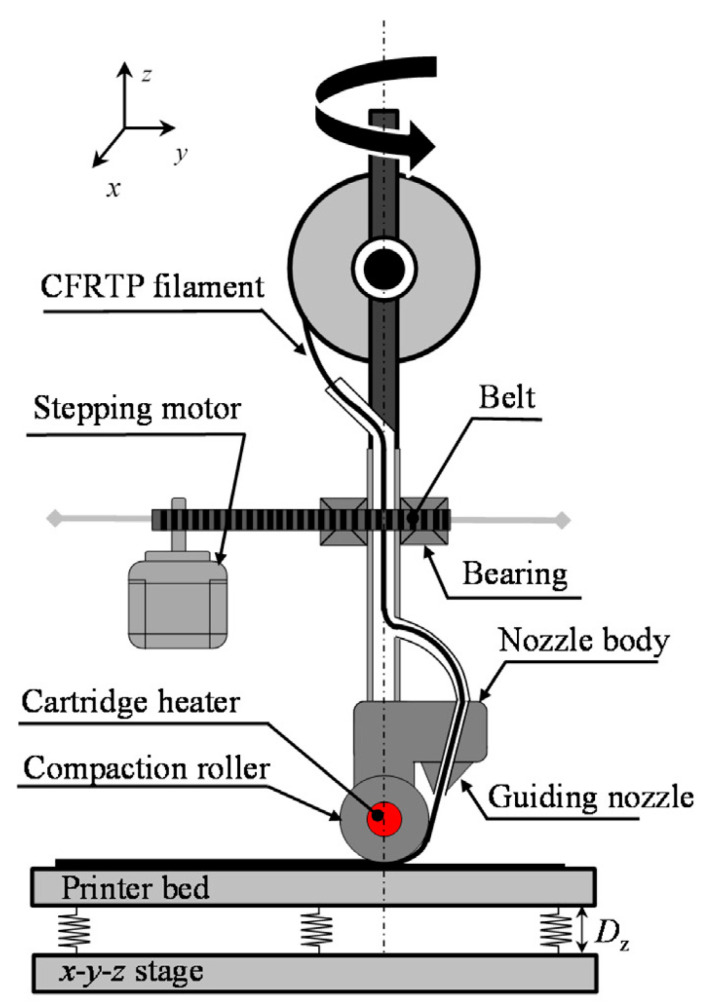
Schematic of 3DCP printer developed by Ueda et al. [129].

**Figure 16 polymers-16-00831-f016:**
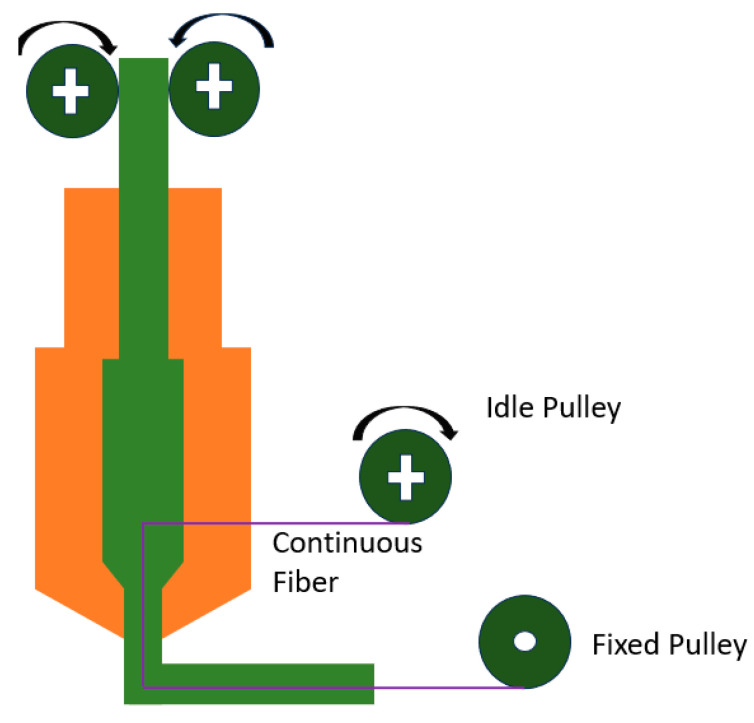
A FDM modified machine developed by Akhundi et al.

**Table 1 polymers-16-00831-t001:** The properties of various composites made with FDM, using different matrix materials and synthetic fibers.

Matrix	Fiber	Content (Volume Fraction)	Results/Highlights	Ref.
ABS	Carbon	6.5	Increasing the flexural strength to 127 MPa, UTS to 147 MPa and decreasing the shear strength to 2.81 MPa compared to ABS processed by injection molding	[92]
ABS	Carbon	1.6	Enhancing the tensile and fatigue strength of fiber reinforced composites with thermal bonding	[93]
Nylon	Carbon Glass Kevlar	26.8 and 72.4 27.5 and 73.8 27.2 and 73.4	The highest shear strength for carbon fiber, glass and Kevlar respectively The improvement of shear strength increases with the increase in fiber volume percentage	[94]
Nylon	Kevlar	4.04, 8.08 and 10.1	Obtaining the elastic modulus of 1767, 6920 and 9001 for three reinforced composites with different volume percentages	[6]
PLA	Carbon	6.6	Presenting and developing a new method of impregnation continuous fiber inside the filament and simultaneous printing	[36]
PLA	Carbon	27	Using continuous fiber impregnation in filament and achieving bending strength and bending modulus of 335 MPa and 30 GPa	[92]
PLA	Carbon	34	Continuous fiber surface preparation to strengthen matrix and fiber adhesion The increase in tensile and bending strength of the modified composite was found by 14 and 164% compared to the unprocessed fiber reinforced composite	[95]
PLA	Aramid	8.6	Comprehensive investigation of mechanical properties for reinforced composite and comparison with PLA	[96]
PLA	Carbon Flax	18.86 and 24.04 9.82, 24.54, 29.45 and 39.27	430% and 325% increase in tensile strength for reinforced composites with carbon fiber and flax fibers, respectively	[97]
TPU PLA PLA-Wood HD PA POM	Glass	34.8 30.5 33.6 31.3 36.3 37.5	Presenting a new method called in-melt simultaneous impregnation and increasing the tensile strength and elastic modulus by more than 700%.	[98]
PETG	Aramid	45	The tensile modulus and strength in the fiber direction increase linearly with fiber loading, resulting in a significantly higher modulus (+1550%) compared to non-reinforced 3D-printed PETG reference materials, as well as a moderately increased strength (+1150%). However, tensile strength perpendicular to the fiber direction experiences a significant decline compared to the reference materials. This decline is attributed to imperfect fiber impregnation and a lack of optimized fiber sizing for the aramid/PETG interface. Additionally, flexural modulus and strength also increase linearly with fiber loading, reaching up to +1650% and +490%, respectively.	[99]
PETG	20		The tensile test results of the 3D-printed PETG/CF solid structural design revealed a 23% improvement in yield strength compared to other conventional structures.	[100]

**Table 2 polymers-16-00831-t002:** Various composites’ properties made with FDM, using different matrix materials and natural fibers.

Matrix	Fiber	Results/Highlights	Ref.
PLA	Flax	The tensile modulus and strength values increased. The tensile properties were in the same range as those for continuous glass fiber/polyamide (PA) printed composites. However, their weakest point was their transverse properties, which remained poorer than similar flax/PLA thermocompressed composites.	[117]
PLA	Flax	The flexural strength and modulus of the 3D-printed flax-reinforced PLA specimens increased by 211% and 224%, respectively, compared with PLA specimens. The maximum bending force load and stiffness of the 3D-printed composite increased by 39% and 115%, respectively.	[53]
PLA	Cotton	Cotton fiber-reinforced composites have shown exceptional tensile strength and stiffness, allowing them to rival synthetic fibers like glass-reinforced composites.	[118]
ABS	Kenaf	The tensile and flexural tests revealed a decrease in the tensile strength and modulus of kenaf fiber-reinforced ABS (KRABS) composites from 0 to 5% kenaf fiber content, which were 23.20 to 11.48 MPa and 328.17 to 184.48 MPa, respectively. Increasing the kenaf fiber content to 5–10% resulted in an increase in tensile strength and modulus from 11.48 to 18.59 MPa and 184.48 to 275.58 MPa, respectively. The flexural strength and modulus of KRABS composites decreased from 40.56 to 26.48 MPa and 113.05 to 60 MPa at 5% kenaf fiber content. Further addition of kenaf fiber from 5 to 10% increased the flexural strength and modulus from 26.48 to 32.64 MPa and 60 to 88.46 MPa, respectively.	[119]
PBS	Hemp	The Young’s modulus of PBS can be improved by 63% by introducing hemp fibers in conjunction with overlap. In contrast, hemp fiber reinforcement reduces the tensile strength of PBS, but this effect is less pronounced when considering overlap in the additive manufacturing process.	[120]
PP	Hemp	The results showed that the 5% hemp PP composite exhibited the highest tensile strength, while the 20% hemp PP composite showed the highest Young’s modulus. These results emphasize the importance of hemp fiber content in altering the mechanical properties of a polymeric material to achieve the desired properties for specific industry needs.	[121]
PLA	Wood	The experimental results indicated that aligning wood fibers within PLA polymer resulted in enhanced mechanical performance.	[122]
PLA	Basalt	The results suggest that PLA/KBF exhibits comparable tensile properties and superior flexural properties compared to the PLA/CF control. This can be attributed to the high complex viscosity of PLA/CF, which affects interlayer adhesion.	[123]

**Table 3 polymers-16-00831-t003:** A summary of research based on 3D printing of reinforced composites using dual extruder FDM mechanism.

Matrix Material	Reinforcement Filament (s)	FDM Printing Pattern	Tests (Tested Properties)	Ref.
Ultem	Printable CNT yarn filaments (The average diameter of the filaments containing 10–30% resin by weight was around 350 μm)	unidirectional layup pattern	Mechanical and Electrical properties. Tensile test, material characterization tests, electrical conductivity tests.	[26]
PA6 In 1.75 mm diameter	sized car bon fiber (SCF) and virgin carbon fiber (VCF)	-	interfacial performance and fracture patterns, flexural strength and modulus	[127]
PA (Nylon)	Continuous glass or carbon fibers	elliptical patterns	plane strength and stiffness properties of the composites	[128]
ABS	Carbon fiber	The infill pattern deposition directions for different layers were 45 and 135 degrees	Strength, ductility, stiffness, toughness	[75]

**Table 4 polymers-16-00831-t004:** A comparison between different FDM mechanisms.

Mechanism Type of the FDM Process	Advantages	Disadvantages
In-situ Fusion mechanism	The user has control over the flow rate of the thermoplastic content and can adjust it according to the application of the printed part It is a single-step manufacturing method.	The poor bonding between the layers (fiber and matrix) due to the short dwell time, which results in a weakened strength of the printed part The quick dwell time in the nozzle leads to inadequate polymer infusion into the fiber bundles, resulting in increased porosity and weaker mechanical properties
Dual extruder	Have greater flexibility Offers higher precision Enables combination of different filaments and reinforcements Improve mechanical properties	High cost of building and maintenance Needing more filament Setting it up can be Time-consuming
3D compaction printing	Increases the adhesion between deposited layers (beads) Lessens the voids Promotes the mechanical behavior	The printer is not as simple as others. It needs post processing
Modified in-situ fusion mechanism	Capability for online changing of the fiber fraction volume	

## Data Availability

Not applicable.

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
