# Peer review of "Various FDM Mechanisms Used in the Fabrication of Continuous-Fiber Reinforced Composites: A Review"

_polymers, 2024, doi:10.3390/polym16060831_

Round 1

Reviewer 1 Report

Comments and Suggestions for Authors

Dear Editor, in this review the use of Fused Deposition Modeling (FDM), which is an additive manufacturing technology, and mainly its use in the fabrication of continuous-fiber reinforced composites is extensively described. All published works have been discussed and their main findings have been described in a proper way. Authors focused mainly to explain comprehensively two main FDM mechanisms: in-situ fusion and ex-situ prepreg. The review has some merit and deserves to be published. I have pnly few comments.

Please correct the names of all mentioned polymers. Most of them are not well named. For example, polylactic acid should be written poly(lactic acid). A parenthesis is always needed when the polymer name consists from two words. Only in the case that is name with one word like polyethylene or polypropylene, a parenthesis in not needed.

What are the advantages and disadvantages of each one mechanism? This could be added before conclusions.

Comments on the Quality of English Language

Minor editing of English language required

Author Response

Comments and Suggestions for Authors

Dear Editor, in this review the use of Fused Deposition Modeling (FDM), which is an additive manufacturing technology, and mainly its use in the fabrication of continuous-fiber reinforced composites is extensively described. All published works have been discussed and their main findings have been described in a proper way. Authors focused mainly to explain comprehensively two main FDM mechanisms: in-situ fusion and ex-situ prepreg. The review has some merit and deserves to be published. I have only few comments.

Dear Respected Reviewer,

We would like to express our sincere gratitude for your time and effort to review our manuscript and provide insightful feedback and comments that helped us revise the manuscript. We have prepared a revised version enclosed, shared our views with you, and done our best to address your valuable comments and apply your great suggestions. All of them have helped us to improve the quality of the manuscript and we hope that you find this revised version of the manuscript satisfactory. Below you can find our point-by-point responses, and changes highlighted in green.

Please correct the names of all mentioned polymers. Most of them are not well named. For example, polylactic acid should be written poly(lactic acid). A parenthesis is always needed when the polymer name consists from two words. Only in the case that is name with one word like polyethylene or polypropylene, a parenthesis in not needed.

Answer: Thanks for your comment. It has been edited.

What are the advantages and disadvantages of each one mechanism? This could be added before conclusions.

Answer: A table containing advantages and disadvantages of each mechanism has been added before conclusion.

Comments on the Quality of English Language

Minor editing of English language required

Answer: Thanks for your comment. English language was edited.

Reviewer 2 Report

Comments and Suggestions for Authors

Comments

1.     In Figure 1 liquefier is not a correct term. Suggested to change them.

2.     The authors reviewed only one type of natural fiber reinforcement. There are various works on natural fiber reinforcement in FDM. The authors are suggested to provide a separate section.

3.     Mostly the reinforcements in the FDM technique used filler type of reinforcement in the filaments. Hence a section about filler type of reinforcement should be added.

4.     The introduction can be improved by considering https://doi.org/10.1177/00952443241229186, https://doi.org/10.1002/pc.28133.

5.      Synthetic material reinforced filament is a common trend. Suggested to make a table similar to table 1 for the natural materials reinforced filaments for printing.

6.     The manuscript lacks review points over the short fiber reinforced filaments.

7.     The authors must provide insights about how these filaments are produced and how they are reinforced with reinforcements.

8.     The structure must be corrected with a flow-distinguishing filler, short fiber, and continuous fiber reinforcement.

9.     Provide a data table about the different types of 3D printing filament material. The review manuscript only highlights a few kinds.

10.  Over all the manuscript is good, but needs the following revisions before further processing.

Author Response

Comments and Suggestions for Authors

  1. In Figure 1 liquefier is not a correct term. Suggested to change them.

Answer: Figure 1 has been corrected and the term "liquifier" has been changed to "heater".

  1. The authors reviewed only one type of natural fiber reinforcement. There are various works on natural fiber reinforcement in FDM. The authors are suggested to provide a separate section.

Answer: Thank you for the suggestion. A separate section was added about natural fibers.

  1. Mostly the reinforcements in the FDM technique used filler type of reinforcement in the filaments. Hence a section about filler type of reinforcement should be added.

Answer: Although the article is mainly about continuous fibers in FDM and not fillers, a section regarding filler types of reinforcements has been added based on your comment.

  1. The introduction can be improved by considering https://doi.org/10.1177/00952443241229186, https://doi.org/10.1002/pc.28133.

Answer: Thank you for your suggestion. The references have been used in the introduction.

  1. Synthetic material reinforced filament is a common trend. Suggested to make a table similar to table 1 for the natural materials reinforced filaments for printing.

Answer: Thank you for your suggestion. Another table including natural fibers has been added similar to table 1.

  1. The manuscript lacks review points over the short fiber reinforced filaments.

Answer: As the title of the manuscript shows, this is mainly focused on continuous fibers. However, a section related to short fibers has been also put into the article.

  1. The authors must provide insights about how these filaments are produced and how they are reinforced with reinforcements.

Answer: A section related to filaments preparation, their different types, and how they are reinforced is added.

  1. The structure must be corrected with a flow-distinguishing filler, short fiber, and continuous fiber reinforcement.

Answer: The structure of the article has been modified based on your comment.

  1. Provide a data table about the different types of 3D printing filament material. The review manuscript only highlights a few kinds.

Answer: A separate section was added containing information related to filaments.

  1. Over all the manuscript is good, but needs the following revisions before further processing.

Dear Respected Reviewer,

We would like to express our sincere gratitude for your time and effort to review our manuscript and provide insightful feedback and comments that helped us revise the manuscript. We have prepared a revised version enclosed, shared our views with you, and done our best to address your valuable comments and apply your great suggestions. All of them have helped us to improve the quality of the manuscript and we hope that you find this revised version of the manuscript satisfactory. Below you can find our point-by-point responses, and changes highlighted in green.

Reviewer 3 Report

Comments and Suggestions for Authors

The review is not complete and systematic and needs to be significantly structured and corrected.

1) Line 181 – low density is not about glass fibers (2.5 kg/m3) with comparison to carbon (1.8) or aramides

2) Section 3 seems illogical:

Short and long fibers should be considered separately.

Why is subsection 2.2.4.1 needed? if the authors in section 2.2.4 already talk about Kevlar and do not cite other representatives of the aramid class?

There is no section about basalt fibers - they are also used in FDM printing (although they are heavier than fiberglass).

3) Does Table 1 show data for composites with continuous reinforcement or with short fibers?

4) It’s strange not to see an example based on PET-G plastic in the table. Anisoprint is engaged in the industrial production of printers for this technology.

5) One of the main shortcomings of the review is the lack of information about highly filled prepregs obtained by the Anisoprint company (more than 90% filling), fiber filling with thermosets (Although Anisoprint is not the only example).

https://scholar.google.com/scholar?hl=en&as_sdt=0%2C5&q=anisoprint

6) One of the main shortcomings of the review is the lack of information about Markforged technology. It is not recommended to indicate specific links in reviews, so here are just results for the keyword; authors should study this topic in more detail: https://scholar.google.com/scholar?hl=en&as_sdt=0%2C5&q=markforged.

7) At the same time, the work contains a description of many unimportant details, for example, a detailed indication of a series of samples and their parameters on page 15.

8) Lines 11-13.

In my opinion, the reason for the low strength is not “polymer characteristics”. The main reasons for the low strength of products produced by FDM 3D printing are high porosity and low sinterability of monofilaments (layers).

9) Fig 2.

Some points, such as safety, the potential of using new materials, the potential of using various engineering plastics, and also: materials in filament form, limited accuracy, are controversial. Nothing is said about the cost, limitation of product size, ease of maintenance of the 3D printer, shrinkage.

10) Lines 65-66

Are you comparing to carbon fibers? then the strength is really low.

11) Lines 77-79 Here is the same written twice: “Due to their different orientations, composites fabricated with continuous fibers exhibit higher strength than those with short fibers, thanks to their orientation.”

12) Line 109 polyamide (PA) [47, 48], Nylon[49–51],

Nylon is a PA class polymer (as well as aramides)

13) There are four times in the text written the same phrase:

Lines 1314

 Regarding polymer characteristics, there are two main types of reinforcing fibers: discontinuous (short) and continuous.”

Lines 74-75 

These reinforcing materials can be divided into continuous and discontinuous (short) fibers [33].

Lines 117- 118

Another part of an FRC is reinforcing composite, which is introduced to increase the 117 composite strength and divided into discontinuous (short) and continuous fibers

Lines 145- 146

There are two types of CFRPs: discontinuous or short CFRPs, and continuous CFRPs [18, 54, 55].”

14) Lines 160-162 Which are studies? There are no refs.

15) Lines 174-175 Here is incorrect written sentence

16) Lines 191-193 Here is incorrect written sentence

17) Lines 370-373 Here the second sentence means the same as the first.

18) 2 paragraphs from 527 to 534 lines and from 542 to 549 lines are simply the same. Lines 542 to 549 were inserted there by mistake.

19) There is no 3D printing scheme with 2 extruder heads (Dual Extruder Mechanism/ Ex-Situ Method).

Minor flaws:

The article contains many typos and inaccuracies and requires proofreading:

Line 482 here should be “Type B”, 15 (a) (figure numbers should be in parentheses), etc., 0.2 mm (there should be a space between the number and dimension). In addition, there are a lot of double spaces, missing dots, units of measurement with a small letter (Mpa), etc. (there are dozens of such errors in the Manuscript)

Comments on the Quality of English Language

A number of inappropriate proposals, some indicated in the main review

Author Response

Comments and Suggestions for Authors

The review is not complete and systematic and needs to be significantly structured and corrected. 

Dear Respected Reviewer,

We would like to express our sincere gratitude for your time and effort to review our manuscript and provide insightful feedback and comments that helped us revise the manuscript. We have prepared a revised version enclosed, shared our views with you, and done our best to address your valuable comments and apply your great suggestions. All of them have helped us to improve the quality of the manuscript and we hope that you find this revised version of the manuscript satisfactory. Below you can find our point-by-point responses, and changes highlighted in green.

1) Line 181 – low density is not about glass fibers (2.5 kg/m3) with comparison to carbon (1.8) or aramides.

Answer: The sub-section related to glass fibers has been modified.

2) Section 3 seems illogical: Short and long fibers should be considered separately.

Answer: Thanks to your comment. Short and continuous fibers are now put into two different sections.

Why is subsection 2.2.4.1 needed? if the authors in section 2.2.4 already talk about Kevlar and do not cite other representatives of the aramid class?

There is no section about basalt fibers - they are also used in FDM printing (although they are heavier than fiberglass).

Answer: A section related to basalt fibers is added.

3) Does Table 1 show data for composites with continuous reinforcement or with short fibers?

Answer: The main focus of this article is continuous fibers, therefore, most of the information in the table are related to continuous fibers.

4) It’s strange not to see an example based on PET-G plastic in the table. Anisoprint is engaged in the industrial production of printers for this technology.

Answer: Information related to PETG was added to the table1.

5) One of the main shortcomings of the review is the lack of information about highly filled prepregs obtained by the Anisoprint company (more than 90% filling), fiber filling with thermosets (Although Anisoprint is not the only example).

https://scholar.google.com/scholar?hl=en&as_sdt=0%2C5&q=anisoprint

Answer: A paragraph was added related to Markforged and Anisoprint technologies.

6) One of the main shortcomings of the review is the lack of information about Markforged technology. It is not recommended to indicate specific links in reviews, so here are just results for the keyword; authors should study this topic in more detail: https://scholar.google.com/scholar?hl=en&as_sdt=0%2C5&q=markforged.

 Answer: A paragraph was added related to Markforged and Anisoprint technologies.

7) At the same time, the work contains a description of many unimportant details, for example, a detailed indication of a series of samples and their parameters on page 15.

Answer: Unnecessary information related to series of samples and their parameters was removed.

8) Lines 11-13.

In my opinion, the reason for the low strength is not “polymer characteristics”. The main reasons for the low strength of products produced by FDM 3D printing are high porosity and low sinterability of monofilaments (layers).

Answer: The sentence was modified.

9) Fig 2.

Some points, such as safety, the potential of using new materials, the potential of using various engineering plastics, and also: materials in filament form, limited accuracy, are controversial. Nothing is said about the cost, limitation of product size, ease of maintenance of the 3D printer, shrinkage.

Answer: The figure is now modified.

10) Lines 65-66

Are you comparing to carbon fibers? then the strength is really low.

Answer: To avoid confusion the sentence was edited.

11) Lines 77-79 Here is the same written twice: “Due to their different orientations, composites fabricated with continuous fibers exhibit higher strength than those with short fibers, thanks to their orientation.”

Answer: It has been corrected.

12) Line 109 polyamide (PA) [47, 48], Nylon[49–51],

Nylon is a PA class polymer (as well as aramids)

Answer: Thanks for your consideration. It has been corrected.

13) There are four times in the text written the same phrase:

Lines 13–14

 “Regarding polymer characteristics, there are two main types of reinforcing fibers: discontinuous (short) and continuous.”

Lines 74-75 

These reinforcing materials can be divided into continuous and discontinuous (short) fibers [33].

Lines 117- 118

Another part of an FRC is reinforcing composite, which is introduced to increase the 117 composite strength and divided into discontinuous (short) and continuous fibers

Lines 145- 146

There are two types of CFRPs: discontinuous or short CFRPs, and continuous CFRPs [18, 54, 55].”

Answer: Thanks to your comment. Repeated sentences were removed.

14) Lines 160-162 Which are studies? There are no refs.

Answer: References were added.

15) Lines 174-175 Here is incorrect written sentence

Answer: Sentences were corrected.

16) Lines 191-193 Here is incorrect written sentence

Answer: It has been corrected.

17) Lines 370-373 Here the second sentence means the same as the first.

Answer: The second sentence was removed.

18) 2 paragraphs from 527 to 534 lines and from 542 to 549 lines are simply the same. Lines 542 to 549 were inserted there by mistake.

Answer: The second paragraph was removed.

19) There is no 3D printing scheme with 2 extruder heads (Dual Extruder Mechanism/ Ex-Situ Method).

Answer: A schematic related to 2-extruder mechanism has been added.

Minor flaws:

The article contains many typos and inaccuracies and requires proofreading:

Line 482 here should be “Type B”, 15 (a) (figure numbers should be in parentheses), etc., 0.2 mm (there should be a space between the number and dimension). In addition, there are a lot of double spaces, missing dots, units of measurement with a small letter (Mpa), etc. (there are dozens of such errors in the Manuscript)

Answer: Errors have been edited.

Comments on the Quality of English Language

A number of inappropriate proposals, some indicated in the main review.

Answer: Thanks for your comment. English language was edited.

Round 2

Reviewer 2 Report

Comments and Suggestions for Authors

All the comments were addressed and the manuscript can be accepted in its current form.

Author Response

Comments and Suggestions for Authors

Reviewer 2:

All the comments were addressed and the manuscript can be accepted in its current form.

Dear Respected Reviewer,

We would like to express our sincere gratitude for your time and effort to review our manuscript and provide insightful feedback and comments that helped us revise the manuscript.

Reviewer 3 Report

Comments and Suggestions for Authors

The content of the work has become much better.

In terms of the Manuscript format, Authors need to carefully check: there are extra brackets somewhere, there are no extra paragraphs (or, on the contrary, they are made); 3.2.3.3. Aramid (line 318) then 2.2.1.4. Kevlar  (line 333) etc. 

Author Response

Comments and Suggestions for Authors

Reviewer 3:

The content of the work has become much better.

In terms of the Manuscript format, Authors need to carefully check: there are extra brackets somewhere, there are no extra paragraphs (or, on the contrary, they are made); 3.2.3.3. Aramid (line 318) then 2.2.1.4. Kevlar  (line 333) etc. 

Dear Respected Reviewer,

We would like to express our sincere gratitude for your time and effort to review our manuscript and provide insightful feedback and comments that helped us revise the manuscript.

Thanks for your suggestion. The manuscript format was checked and the mentioned items were corrected.
